# Efficacy and Safety of PD-1/PD-L1 Inhibitor as Single-Agent Immunotherapy in Endometrial Cancer: A Systematic Review and Meta-Analysis

**DOI:** 10.3390/cancers15164032

**Published:** 2023-08-09

**Authors:** Mohd Nazzary Mamat @ Yusof, Kah Teik Chew, Abdul Muzhill Hannaan Abdul Hafizz, Siti Hajar Abd Azman, Wira Sofran Ab Razak, Muhammad Rafi’uddin Hamizan, Nirmala Chandralega Kampan, Mohamad Nasir Shafiee

**Affiliations:** 1Gynaecologic-Oncology Unit, Department of Obstetrics and Gynaecology, Hospital Canselor Tuanku Muhriz, Universiti Kebangsaan Malaysia, Cheras, Kuala Lumpur 56000, Malaysia; 2Department of Obstetrics and Gynaecology, Faculty of Medicine, Universiti Kebangsaan Malaysia, Cheras, Kuala Lumpur 56000, Malaysia

**Keywords:** endometrial cancer, programmed cell death protein 1, programmed cell death ligand 1, programmed cell death ligand 2, immune checkpoint inhibitor, mismatch repair, clinical trials efficacy, adverse events

## Abstract

**Simple Summary:**

The programmed cell death protein 1 (PD-1)/programmed cell death ligand 1 (PD-L1) pathway controls the induction and maintenance of immune tolerance within the tumor microenvironment. Hence, immunotherapy is an exciting approach, and tremendous strides have recently been made in our perception of the role of the host immune response in affecting tumor growth and response to various therapies, including in endometrial cancer. Clinical trials investigating PD-1/PD-L1 inhibitors have shown promising results in other cancers, but their efficacy in EC remains uncertain, and guidelines must be consistent. Therefore, this meta-analysis aims to provide a robust analysis of the effectiveness and safety of PD-1/PDL1 inhibitor as single-agent immunotherapy in EC, focusing on three major components: objective response rate (ORR), disease control rate (DCR), and adverse events (AEs). Based on the subgroup analysis of mismatch repair (MMR) status, PD-1/PD-L1 inhibitor immunotherapy showed significantly better efficacy in MMR-deficient patients, in terms of both ORR and DCR.

**Abstract:**

The programmed cell death protein 1 (PD-1)/programmed cell death ligand 1 (PD-L1) pathway plays a crucial role in the immune escape mechanism and growth of cancer cells in endometrial cancer (EC). Clinical trials investigating PD-1/PD-L1 inhibitor have shown promising results in other cancers, but their efficacy in EC still remains uncertain. Therefore, this meta-analysis aims to provide an updated and robust analysis of the effectiveness and safety of PD-1/PDL1 inhibitor as single-agent immunotherapy in EC, focusing on the objective response rate (ORR), disease control rate (DCR), and adverse events (AEs). This meta-analysis utilized STATA version 17 and RevMan version 5.4 software to pool the results of relevant studies. Five studies conducted between 2017 and 2022, comprising a total of 480 EC patients enrolled for PD-1/PD-L1 inhibitor immunotherapy met the inclusion criteria. The pooled proportion of EC patients who achieved ORR through PD-1/PD-L1 inhibitor treatment was 26.0% (95% CI: 16.0–36.0%; *p* < 0.05). Subgroup analysis based on mismatch repair (MMR) status showed an ORR of 44.0% (95% CI: 38.0–50.0%; *p* = 0.32) for the deficient mismatch repair (dMMR) group and 8.0% (95% CI: 0.0–16.0%; *p* = 0.07) for the proficient mismatch repair (pMMR) group. Pooled proportion analysis by DCR demonstrated an odds ratio (OR) of 41.0% (95% CI: 36.0–46.0%, *p* = 0.83) for patients undergoing PD-1/PD-L1 inhibitor treatment. Subgroup analysis based on MMR status revealed DCR of 54.0% (95% CI: 47.0–62.0%; *p* = 0.83) for the dMMR group, and 31.0% (95% CI: 25.0–39.0%; *p* = 0.14) for the pMMR group. The efficacy of PD-1/PD-L1 inhibitors was significantly higher in the dMMR group compared to the pMMR group, in terms of both ORR (OR = 6.30; 95% CI = 3.60–11.03; *p* < 0.05) and DCR (OR = 2.57; 95% CI = 1.66–3.99; *p* < 0.05). In terms of safety issues, the pooled proportion of patients experiencing at least one adverse event was 69.0% (95% CI: 65.0–73.0%; *p* > 0.05), with grade three or higher AEs occurring in 16.0% of cases (95% CI: 12.0–19.0%; *p* > 0.05). Based on the subgroup analysis of MMR status, PD-1/PD-L1 inhibitor immunotherapy showed significantly better efficacy among dMMR patients. These findings suggest that patients with dMMR status may be more suitable for this treatment approach. However, further research on PD-1/PD-L1 inhibitor immunotherapy strategies is needed to fully explore their potential and improve treatment outcomes in EC.

## 1. Introduction

According to the Surveillance, Epidemiology and End Results (SEER) program in the United States, the incidence of EC in 2023 is estimated to be 66,200 new cases, with 13,030 deaths among women [1]. This represents a 0.4% increase in new cases and a 3.7% increase in deaths compared to the data from the previous year [2]. The overall five-year EC survival rate was reported to be 81% in 2019, whereas for the disease’s advanced stages, a low survival rate of only 18.4% was recorded [1]. Hence, research on molecular pathogenesis and immune modulation has been conducted to formulate novel targeted therapies, as current treatments and adjuvant therapy have had limited success in improving the survival rate.

Currently, immune checkpoint inhibitors (ICIs) are actively being explored in EC due to their promising results as targeted therapy for other cancers, such melanoma [3]. Among the immune-related markers, PD-1 and PD-L1 have shown remarkable potential as diagnostic tools, targeted therapies, and predictive markers [4]. According to the guidelines of the British Gynaecological Cancer Society (BGCS), published in 2022, all EC cases should undergo immunohistochemical examination for MMR, p53, and estrogen receptor (ER). However, next-generation sequencing for POLE mutations should only be performed in cases where abnormal MMR and/or p53 are detected [5].

To date, the US Food and Drug Administration (FDA) has approved two PD-1/PD-L1 inhibitors (dostarlimab and pembrolizumab) for the treatment of recurrent or advanced EC. These drugs work by reactivating T-cell-mediated antitumor immunity through the inhibition of the PD-1 or PD-L1 immune checkpoint pathways. However, these ICIs have been associated with autoimmune-like disorders due to their reactivation of cellular immunity [6,7,8]. Clinical trials of PD-1 and PD-L1 inhibitors investigating the efficacy and treatment-related adverse events have been performed following the standard guidelines. However, there are significant variations in the published studies regarding the type of cancer, the specific drug used, the dosage schedule, and the criteria for reporting adverse events. Ignoring these differences and failing to consider patterns in data can result in inaccurate calculations of the actual occurrences of PD-1 and PD-L1 inhibitor efficacy and associated adverse events.

We conducted this meta-analysis to evaluate the effectiveness of PD-1/PD-L1 inhibitors as single-agent immunotherapy in EC. We utilized data from current clinical trials to perform a comprehensive analysis, pooling and quantifying the efficacy of treatment based on ORR, DCR, and AEs in patients with dMMR and pMMR EC, respectively.

## 2. Materials and Methods

### 2.1. Protocol and Eligibility Criteria

This systematic review and meta-analysis adhered to the standard guideline protocol stipulated in the preferred reporting items for systematic reviews and meta-analysis (PRISMA) [9]. Furthermore, it was registered in the PROSPERO database with registration number CRD42022360739.

The inclusion criteria for studies in this meta-analysis focused on PD-1/PD-L1 inhibitor as single agent immunotherapy in EC were as follows: The included studies should have adequate clinical information, including MMR status, the name of the drug used, the total number of patients participating, study phases, ORR, DCR, occurrence of adverse events, progression-free survival (PFS), and overall survival (OS), and should be written in English. Studies that were classified as review papers, books, practice guidelines, letters, editorials, commentaries, case reports, pilot studies, and case studies with fewer than ten patients were all excluded from the analysis.

### 2.2. Information Sources and Search Criteria

The search for relevant studies was conducted in the PubMed, Web of Science, and Cochrane Library databases. The PICO tool was used to generate the main search keywords, focusing on the Population, Intervention, Comparison, and Outcomes [10]. Women or patients with EC were the populations of interest. The intervention was defined as PD-1/PD-L1 inhibitor immunotherapy in EC. The comparison was defined as MMR status of EC patients. The outcome measures were determined with reference to clinical trial primary and secondary endpoints. The literature was searched using the keywords (“PD-1 inhibitor” OR “PD-L1 inhibitor” OR “immunotherapy” OR “immune checkpoint inhibitor” OR “atezolizumab” OR “avelumab” OR “dostarlimab” OR “durvalumab” OR “nivolumab” OR “pembrolizumab”) AND (“endometrial cancer” OR “endometrial carcinoma” OR “endometrium cancer” OR “endometrial neoplasms”). The literature search was limited to articles published within a ten-year timeframe, specifically from 2012 to 2022. The search was conducted on 31 December 2022.

### 2.3. Study Selection

The preliminarily selected studies were assessed for their validity and relevance according to the inclusion criteria. Individual study was evaluated, focusing on the information provided, selection bias, and the quality of data analysis. The titles and abstracts of the studies were screened to identify the relevant information. Then, unrelated studies were excluded. The full texts of the selected studies that met the eligibility criteria were thoroughly analyzed. The studies included in the final analysis were assessed for risk of bias. Data extraction was then conducted for the relevant studies.

### 2.4. Data Collection Process and Data Items

Open-access and restricted-access studies were independently reviewed by two authors (M.N.M.@Y. and K.T.C.). If discrepancies arose during the analysis, the third author, M.N.S., was involved in order to resolve them. The UKM library provided institutional support for accessing the literatures. The identified studies were organized using Mendeley to remove duplicate studies. Data were extracted and recorded in a dedicated table designed specifically for this review. These tables included information such as author names, publication year, country, study registration number, MMR status, drug name, study phase, ORR, DCR, number of patients, occurrence of adverse events, median PFS, and median overall OS.

### 2.5. Assessment of the Risk of Bias of Individual Studies

The quality of eligible studies was assessed using the Newcastle–Ottawa Scale (NOS) for cohort studies [11]. The NOS consists of 9 stars, which are divided into three categories: selection (4 stars), comparability (2 stars), and outcome (3 stars). Studies are assigned an NOS score ranging from 0 to 9, indicating their quality. Studies with an NOS score of 6 or higher are considered to be of high quality, with low publication bias.

### 2.6. Statistical Methods

The meta-analysis of proportions and associations was conducted using STATA (version 17, College Station, TX, USA) for pooled effect estimates. The RevMan software package (version 5.4, London, UK) was used for analysis of OR. The effect of immunotherapy was estimated using effect size (ES) or OR, with a corresponding 95% CI, for each study. Pooled proportion or association was reported as 95% CI, and statistical heterogeneity was assessed using Cochrane Q statistic and I-squared (I^2^) statistics. Heterogeneity was considered to be statistically significant if the *p*-value of the Cochrane-Q test was less than 0.05, and the I^2^ statistic value greater than 50%. The fixed-effect model (FEM) was used for pooling analysis in cases where I^2^ had a low heterogeneity. If substantial heterogeneity was observed for I^2^, either the random-effects model (REM) or the quality-effects model (QEM) was utilized for pooling analysis.

## 3. Results

### 3.1. Search Sequence and Quality Assessment of Selected Publications

A total of 732 study records were identified through database searches, and after removing duplicates, 380 records were excluded (Figure 1). The titles and abstracts of the remaining 352 records were screened, leading to the exclusion of an additional 64 studies. The full texts of the remaining 288 studies were thoroughly examined for relevance. Of these, 283 studies were excluded on the basis of the specified criteria (Figure 1). The remaining five studies were all assessed for risk of bias using the Newcastle–Ottawa Scale (NOS), and the results are presented in Table 1. All included studies received NOS scores ranging from 6 to 9, indicating a low risk of bias.

### 3.2. Study Characteristics

A total of five studies, involving 480 EC patients, were finally included in this review [13,14,15,16,17]. The earliest study was conducted by Ott et al. in 2017 [17], while the most recent studies were published in 2022 by Oaknin et al. [15] and O’Malley et al. [16]. Among the included studies, the study by Oaknin et al. was the most comprehensive [15]. One study was a phase I clinical trial, one was a phase Ib trial, and the other three studies were phase II trials. All of the selected studies had been registered on the http://www.clinicaltrials.gov (accessed on 31 January 2023) website. All included studies and variables of interest associated with PD-1/PD-L1 inhibitor immunotherapy in EC patients are summarized in Table 2.

### 3.3. Quantitative Synthesis

#### 3.3.1. Efficacy of PD-1/PD-L1 Inhibitor Immunotherapy in Endometrial Cancer Based on Objective Response Rate (ORR)

##### Overall ORR for PD-1/PD-L1 Inhibitor Immunotherapy in Endometrial Cancer

The meta-analysis of the pooled proportion of EC patients who received PD-1/PD-L1 inhibitor as single-agent immunotherapy resulted in an ORR of 26.0% (95% CI: 16.0–36.0%; *p* < 0.05). However, significant heterogeneity was observed in the pooled proportion of ORR among the EC patients, with the *p*-value of the Cochrane Q statistic being less than 0.05 and the I^2^ statistic value being 81.6%. Considering this substantial heterogeneity, QEM was used to pool the prevalence. Among the included studies, Ott et al. (2017) reported the lowest ORR (13.0%), while O’Malley et al. (2022) reported the highest ORR (48.0%) (Figure 2A).

##### Subgroup Analysis of ORR for PD-1/PD-L1 Inhibitor Immunotherapy in Endometrial Cancer

Subgroup analysis was conducted on the basis of the MMR status of EC patients, specifically distinguishing between dMMR and pMMR status. The pooled proportion of ORR for EC patients with dMMR was found to be 44.0% (95% CI: 38.0–50.0%; *p* = 0.32). FEM was used to pool the prevalence due to there being no significant heterogeneity (*p* = 0.32; I^2^ = 13.8%) observed among the studies (Figure 2B). The pooled proportion of ORR for EC patients with pMMR was 8.0% (95% CI: 0.0–16.0%; *p* = 0.07). FEM was utilized to pool the prevalence, considering that no significant heterogeneity was observed (*p* = 0.07; I^2^ = 63.3%) among the studies (Figure 2C).

Meta-analysis association between MMR status (dMMR and pMMR) and ORR of PD-1/PD-L1 inhibitor immunotherapy was analyzed using the available data in three studies. The efficacy of PD-1/PD-L1 inhibitors was significantly higher in patients with dMMR compared to those with pMMR (OR = 6.30; 95% CI = 3.60–11.03; *p* < 0.05). FEM was used for this analysis, as no significant heterogeneity was observed among the studies (*p* = 0.25; I^2^ = 28%) (Figure 3).

#### 3.3.2. Efficacy of PD-1/PD-L1 Inhibitor Immunotherapy in Endometrial Cancer Based on the Disease Control Rate (DCR)

##### Overall DCR for PD-1/PD-L1 Inhibitor Immunotherapy in Endometrial Cancer

The pooled proportion of EC patients who received PD-1/PD-L1 inhibitor immunotherapy exhibited a DCR of 41.0% (95% CI: 36.0–46.0%; *p* = 0.45). Heterogeneity was not significant (*p* = 0.45; I^2^ < 0.1%), and therefore FEM was used for the analysis (Figure 4A).

##### Subgroup Analysis of DCR for PD-1/PD-L1 Inhibitor Immunotherapy in Endometrial Cancer

Subgroup analysis of the DCR was performed based on the availability of patients’ MMR status (dMMR and pMMR). The pooled proportion of DCR for EC patients with dMMR was 54.0% (95% CI: 47.0–62.0%; *p* = 0.83). FEM was used, as there was no significant heterogeneity (*p* = 0.83; I^2^ < 0.1%) (Figure 4B). Similarly, the pooled proportion of DCR for EC patients with pMMR was 31.0% (95% CI: 25.0–39.0%; *p* = 0.14). FEM was used, as there was no significant heterogeneity (*p* = 0.14; I^2^ = 48.9%) (Figure 4C).

The association between MMR status (dMMR and pMMR) and the DCR of PD-1/PD-L1 inhibitor immunotherapy was analyzed using the data from three studies. The results demonstrated that the efficacy of PD-1/PD-L1 inhibitors was significantly higher in patients with dMMR compared to those with pMMR (OR = 2.57; 95% CI = 1.66–3.99; *p* < 0.05). FEM was used, as there was no significant heterogeneity (*p* = 0.66; I^2^ < 0.1%) (Figure 5).

#### 3.3.3. Safety of PD-1/PD-L1 Inhibitor Immunotherapy in Endometrial Cancer

##### Overall Incidence of Adverse Events (AEs)

The occurrence of AEs of any grade was analyzed in four studies involving patients undergoing PD-1/PD-L1 inhibitor immunotherapy trials for EC. The overall pooled proportion of AEs in EC patients undergoing immunotherapy was 69.0% (95% CI: 65.0–73.0%). The heterogeneity was not found to be significant, as indicated by the *p*-value of the Cochrane Q statistic being greater than 0.05 and the I^2^ statistic value being 39.4%. Therefore, the FEM was used for the analysis (Figure 6A).

##### Incidence of Grade 3 or Higher AEs

Grade 3 or higher AEs were analyzed in four studies involving patients undergoing PD-1/PD-L1 inhibitor immunotherapy trials for EC. The overall pooled proportion of AEs of grade 3 or higher in EC patients undergoing immunotherapy was 16.0% (95% CI: 12.0–19.0%). Heterogeneity was not found to be significant, as indicated by the *p*-value of the Cochrane Q statistic being greater than 0.05 and the I^2^ statistic value being less than 0.1%. Therefore, FEM was used for the analysis (Figure 6B).

##### Treatment-Related AEs Observed in Patients

Among the 435 patients included in the four studies, treatment-related AEs of any grade were frequently experienced, with frequencies as follows: fatigue (19.77%), nausea (13.33%), diarrhea (13.10%), anemia (11.95%), and hypothyroidism (9.77%). When considering only grade 3 and higher treatment-related AEs, patients had a higher incidence of anemia (1.84%), diarrhea (1.84%), fatigue (0.92%), hyperglycemia (0.92%), increased amylase (0.92%), and increased lipase (0.92%). Table 3 provides the complete list of treatment-related AEs observed in the four studies.

#### 3.3.4. Publication Sensitivity Analysis

A sensitivity analysis was conducted to assess the stability and robustness of the pooled results for the efficacy (ORR, DCR) and safety (AEs) measures. The analysis revealed that the range of variation of the pooled results for ORR sensitivity— the overall proportion of ORR sensitivity—was not significantly reversed. The range of the effect size (ES) with 95% confidence intervals (CI) was [0.26 (0.14–0.34) to (0.13–0.47)], *p* < 0.05, indicating consistent results (Figure 7 and Table 4). Similarly, for the other analyses listed in Table 4, the range of variation of the pooled results fell within the ES range, and the results were statistically significant (*p* < 0.05). However, three analyses showed a significant reversal: the pooled analysis of ORR with pMMR [0.08 (0.05, 0.14) to (−0.03, 0.10)], *p* > 0.05 (Table 3), the pooled association of ORR (dMMR vs. pMMR) [6.30 (2.85, 7.08) to (−20.27, 33.62)], *p* > 0.05, and the pooled association of DCR (dMMR vs. pMMR) [2.57 (1.11, 3.53) to (−0.34, 6.88)], *p* > 0.05 (Table 4). These results suggest that the majority of the pooled results for different outcomes were stable and not significantly influenced by any single study. However, there were significant reversals observed in the analysis of the proportion of ORR with pMMR and the association analysis of MMR status with ORR and DCR.

## 4. Discussion

Tumor immunotherapy has emerged in cancer research as a novel therapeutic tool, since the discovery of its mechanism and role in the last decade [18,19]. The research on ICIs has been focused on PD-L1 and anti-PD-1 inhibitors [20,21]. Hence, many studies have been conducted to refine and validate these emerging ICIs in various cancers, including endometrial cancer. Therefore, this meta-analysis pooled the data on the efficacy and safety of the use of PD-1/PD-L1 inhibitor as single-agent immunotherapy in EC patients with the aim of drawing a definitive conclusion on the basis of the available clinical trials.

In clinical trials, OS is considered the gold standard clinical endpoint for assessing the direct therapeutic benefit. However, in the early phase of clinical trials, ORR and DCR are often used as surrogate clinical endpoints for novel targeted therapies when considering them for accelerated approval [22]. Furthermore, in single-arm clinical trials involving patients with recurrent malignancies and no available treatment options, ORR is commonly used as a reference clinical endpoint [23,24]. ORR is a measure of the proportion of patients who achieve a predefined response to treatment, which is typically the complete disappearance of tumors (complete response) or involves a reduction in tumor size (partial response). It serves as an important indicator of the treatment’s efficacy, and can be used to aid decision making in clinical practice and for regulatory approval. Higher ORR generally indicates greater therapeutic efficacy. However, ORR alone does not provide information about long-term or survival outcomes.

Another important secondary endpoint in assessing cancer therapies is DCR, which is closely linked to ORR. DCR takes into account not only patients who experience complete response and partial response, but also those with stable disease. This is particularly relevant in cases where the primary goal of treatment is to stabilize the disease rather than to achieve complete eradication, which is the case especially in the treatment of recurrent EC. DCR has significant utility, particularly when evaluating therapies that primarily exert tumoristatic effects [24,25]. For instance, DCR was demonstrated to predict subsequent survival in patients with advanced small-cell lung cancer in phase II clinical trials [26].

When evaluating the efficacy of clinical trials, an ORR greater than 30.0% is considered an acceptable endpoint for single-arm trials that aim to demonstrate the breakthrough activity of single-agent cancer therapeutics [23]. In this meta-analysis, the overall pooled proportion of ORR was 26.0%, therefore not exceeding the 30.0% threshold. However, when analyzing the subgroup based on MSI status, the ORR for patients with dMMR was significantly higher, at 44.0%, than that of 8.0% for patients with pMMR. The odds ratio (OR) for achieving the ORR was 6.30-fold higher in the dMMR subgroup compared to the pMMR subgroup. The use of ICIs as a single-agent immunotherapy appears to be a highly effective treatment for patients with dMMR or MSI-high endometrial cancers (ECs). The reported overall tumor response rates (ORRs) in this context ranged from 27% to 57% [27].

In terms of DCR, the overall pooled proportion was 41.0%, indicating that 41.0% of patients achieved disease control. When examining the MSI subgroup, the DCR for patients with dMMR was significantly higher, at 54.0%, compared to 31.0% for patients with pMMR. The OR for achieving a DCR was 2.57-fold higher in the dMMR subgroup than in the pMMR subgroup. It has been reported that approximately 13–30% of recurrent ECs are dMMR [28]. The high DCR demonstrated in this meta-analysis indicates a promising prospect for single-agent immunotherapy using ICIs in the treatment of recurrent EC. The favorable DCR suggests that ICIs have the potential to effectively control the disease and achieve meaningful clinical outcomes in this patient population.

In 2014, the FDA granted approval to pembrolizumab and nivolumab, both immune checkpoint inhibitors, for the treatment of certain cancers. These drugs target the interaction between PD-1 and its ligands PD-L1 and PD-L2. The approvals were based on their ORR, observed in clinical trials [29,30]. Similarly, avelumab and atezolizumab received accelerated approval in 2016 for the treatment of urothelial carcinoma and non-small cell lung cancer (NSCLC), respectively, on the basis of ORR outcomes [31,32]. Furthermore, avelumab and durvalumab were granted accelerated approval in 2017 for the treatment of urothelial cancer, also on the basis of ORR [31].

For EC, dostarlimab was the first ICI to receive accelerated approval, in April 2021, for use in single-agent immunotherapy. It was approved for the treatment of advanced or recurrent endometrial cancer with dMMR [33,34]. Pembrolizumab was granted approval in March 2022, as single agent immunotherapy for EC with dMMR status. These approvals highlight the growing success and recognition of ICIs as effective treatments in various cancer types. The use of ORR as an endpoint for approval signifies the importance of tumor response rates in assessing the efficacy of these drugs. It also underscores the significance of identifying specific molecular characteristics, such as dMMR status, in guiding treatment decisions and providing targeted therapies for patients with EC.

A systematic summary of the included studies revealed that the occurrence of treatment-related AEs (any grade) in EC patients undergoing immunotherapy was 69.0%. However, the occurrence of grade 3 or higher AEs was only 16.0%. Among AEs of any grade, fatigue was the most frequently experienced (19.77%), followed by nausea, diarrhea, anemia, and hypothyroidism. Grade 3 or higher treatment-related AEs were reported with low incidence, and the most frequently experienced were anemia (1.84%), diarrhea (1.84%), fatigue (0.92%), hyperglycemia (0.92%), increased amylase (0.92%), and increased lipase (0.92%). Although adverse events were reported in 60% of cases, grade 1–2 adverse events generally did not result in significant complications for patients. However, the discomfort associated with these events often decreases patients’ motivation to continue therapy [35].

Molecular status is important when arriving at treatment decisions for EC patients, as is evidenced by this review focusing on MMR status and the efficacy of single-agent immunotherapy. Therefore, it is recommended that the current guidelines by the Institute of Medicine (IOM) for the most recently developed model, ProMisE (Proactive Molecular Risk Classifier for Endometrial Cancer), be followed to aid in making molecular-based decisions [36,37]. Firstly, the presence or absence of two mismatch repair (MMR) proteins, mutS homolog 6 (MSH6) and PMS1 homolog 2 mismatch repair system components (PMS2), is determined using immunohistochemistry (IHC) for these proteins. If these two proteins are absent, the EC patient is classified as belonging to the dMMR subgroup. Secondly, if the proteins MSH6 and PMS2 are expressed, further analysis via sequencing is performed to identify POLE exonuclease domain mutation (POLE EDM). If this mutation is present, the patient will be classified as a POLE ultra-mutated group. Lastly, if the patient cannot be classified under dMMR or POLE EDM, the status of p53, wild type, or null/missense mutations will be checked using IHC [36,38,39].

In addition to single-agent immunotherapy, combined therapy has also been actively explored in clinical trials for endometrial cancer (EC). One of the most remarkable combined therapies is Pembrolizumab plus Lenvatinib, for which the FDA has recently accelerated approval for use in EC cases that have not progressed after prior treatment and are not classified as dMMR [36]. Lenvatinib is a multikinase inhibitor that targets various receptors involved in promoting tumor growth and angiogenesis. It acts against vascular endothelial growth factor receptors (VEGFR1, VEGFR2, VEGFR3), fibroblast growth factor receptors (FGFR1–4), KIT proto-oncogene, receptor tyrosine kinase (KIT), rearranged during transfection (RET), and platelet-derived growth factor receptor alpha (PDGFRa), which can lead to immunological activation. Additionally, Lenvatinib inhibits platelet-derived growth factor receptor beta (PDGFRb) [40]. The combination of Pembrolizumab and Lenvatinib aims to enhance the anti-tumor immune response while simultaneously showing anti-angiogenesis effects. This approach may have synergistic effects in slowing tumor growth and improving patient outcomes. However, it is important to note that combined therapy with immunotherapy and targeted agents can lead to increased toxicity and potentially severe adverse events. Therefore, ongoing clinical trials are closely monitoring and evaluating the safety and efficacy of this combination therapy to ensure appropriate management of side effects.

This systematic review and meta-analysis demonstrated a more significant association between efficacy and dMMR compared to pMMR in clinical immunotherapy trials for endometrial cancer patients. The safety profile of these treatments showed a low incidence of grade 3 or higher adverse events, supporting the feasibility of proceeding with subsequent phases of trials. The findings suggest that MMR status can serve as a predictive factor for treatment outcomes, as MMR deficiency was linked to poor survival based on a The Cancer Genome Atlas (TCGA)-based approach [41]. However, solid evidence for the validity of MMR status and PD-1/PD-L1 expression in relation to ICI therapy in EC is still lacking. In certain tumor types, predictive biomarkers have been established as companion platforms for tumor assessment in the context of immune checkpoint inhibitor (ICI) response. Cervical cancer is one example of such a case [27,42,43]. Hence, this systematic review can aid in consolidating the diverse outcomes from individual studies and establishing correlations with results in clinical settings using more standardized and feasible approaches. The MSI status in advanced-stage EC patients should be given priority for screening, and if identified, they should be offered with ICI treatment, as regulatory agencies have already approved pembrolizumab and dostarlimab for this purpose [44].

While this systematic review and meta-analysis provide valuable insights, it is important to acknowledge several limitations that should be considered when interpreting the findings and making recommendations. Firstly, not all included studies reported the MMR status of the patients. Categorizing patients on the basis of their MMR status is crucial, as previous studies have shown significant outcomes in immunotherapeutic clinical trials. In future meta-analyses, it would be beneficial to pool data specifically for patients with MMR status. Secondly, not all of the analyses were subgrouped into PD-L1 or PD-1 antibody inhibitors due to data limitations. For instance, Antill et al., 2021 [13] concluded that the AEs recorded were not related to the drug being tested. Therefore, only one study with an anti-PD-L1 inhibitor remains: that of Konstantinopoulos et al., 2019 [14]. In addition, previous studies have stated that toxicity and efficacy are similar when comparing PD-1 and PD-L1 inhibitors in other cancers, such as non-small cell lung cancer (NSCLC) [45,46]. Therefore, we combined the analysis of PD-1 and PD-L1 inhibitors in this review. In future, when more data become available, a more comprehensive meta-analysis can be performed. Lastly, the median OS and PFS were not available for meta-analysis, likely due to the fact that the median rates were not reached in the included studies.

## 5. Conclusions

This systematic review and meta-analysis provided an overview of the pooled proportion of ORR and DCR when using PD-1/PD-L1 inhibitor for single-agent immunotherapy in patients with EC. EC patients with dMMR had significantly better ORR and DCR compared to patients with pMMR. The ICIs were found to be safe, with grade 3 or greater AEs reported in only 16% of cases. Clinically, these findings can serve as a reference for improving treatment modalities by considering detailed patient profiles. To advance personalized medicine, it is necessary to finalize patient profiles and identify predictive biomarkers, particularly for patients with advanced stages of EC, and especially those with dMMR.

## Figures and Tables

**Figure 1 cancers-15-04032-f001:**
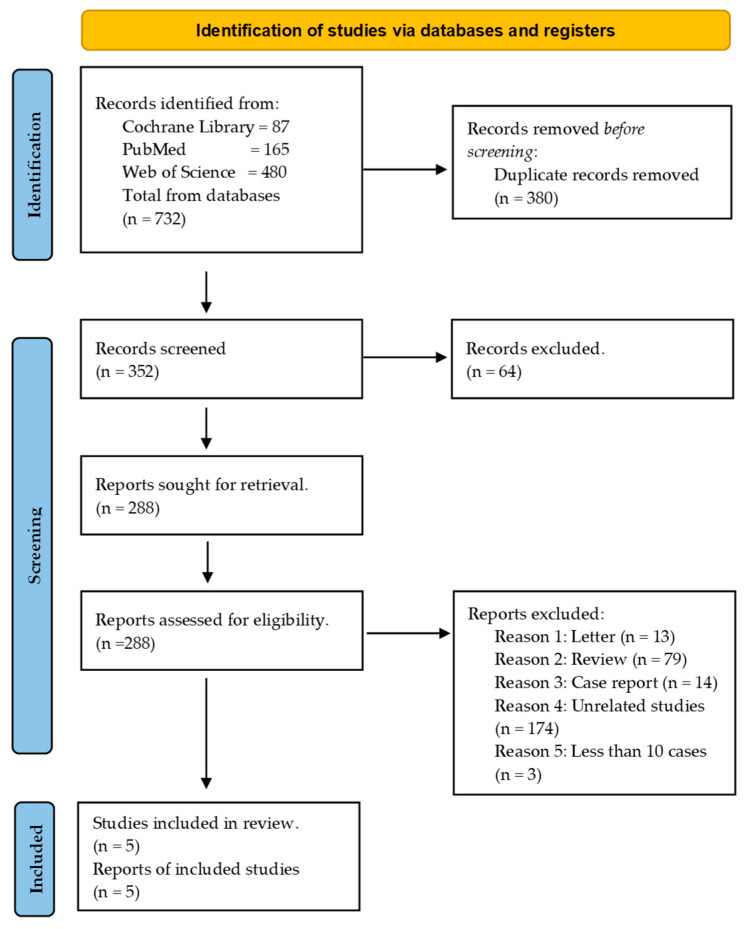
PRISMA flow diagram of study selection and screening [12].

**Figure 2 cancers-15-04032-f002:**
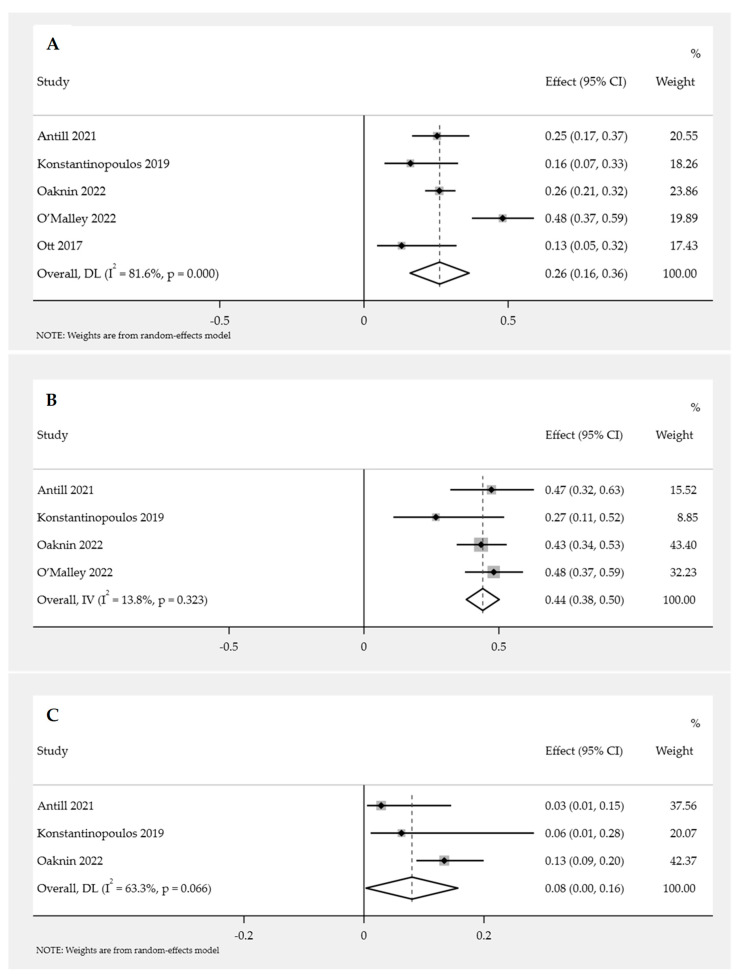
Pooled proportion of ORR for PD-1/PD-L1 inhibitor immunotherapy in endometrial cancer patients [13,14,15,16,17]. Forest plot of (**A**) overall ORR; (**B**) ORR with dMMR; (**C**) ORR with pMMR.

**Figure 3 cancers-15-04032-f003:**
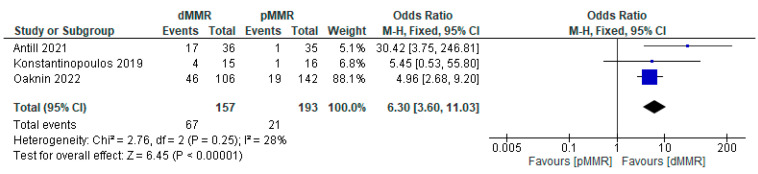
Forest plot of MMR status (dMMR versus pMMR) and ORR for PD-1/PD-L1 inhibitor immunotherapy in endometrial cancer patients [13,14,15].

**Figure 4 cancers-15-04032-f004:**
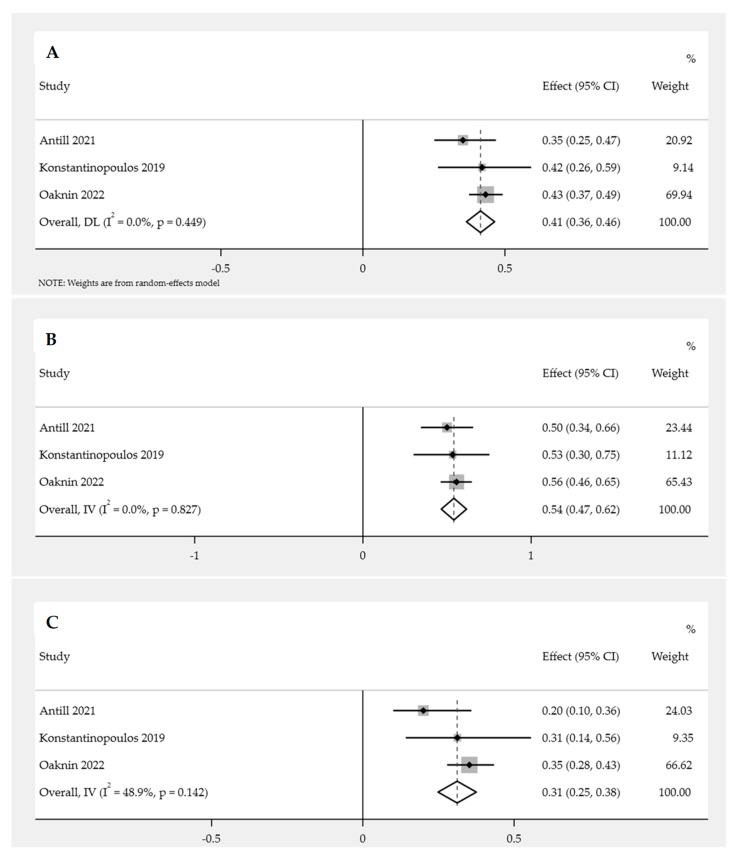
Pooled proportion of DCR for PD-1/PD-L1 inhibitor immunotherapy in endometrial cancer patients [13,14,15]. Forest plot of (**A**) overall DCR; (**B**) DCR with dMMR; (**C**) DCR with pMMR.

**Figure 5 cancers-15-04032-f005:**
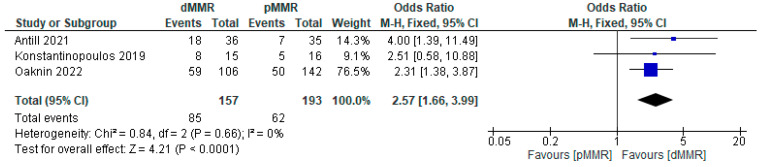
Forest plot of MMR status (dMMR versus pMMR) and DCR for PD-1/PD-L1 inhibitor immunotherapy in endometrial cancer patients [13,14,15].

**Figure 6 cancers-15-04032-f006:**
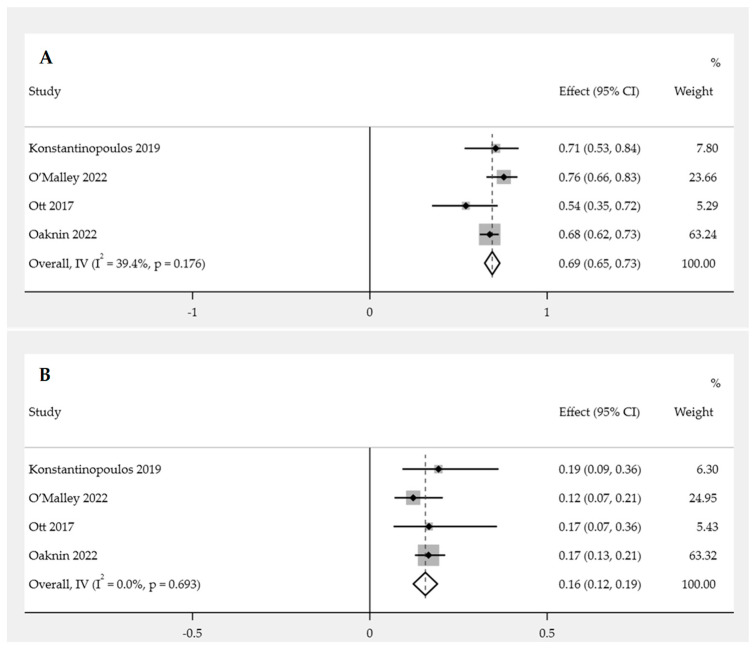
Pooled proportion of adverse event incidence for PD-1/PD-L1 inhibitor immunotherapy in endometrial cancer patients [14,15,16,17]. Forest plot of (**A**) overall incidence of adverse events; (**B**) incidence of adverse events of grade 3 or higher.

**Figure 7 cancers-15-04032-f007:**
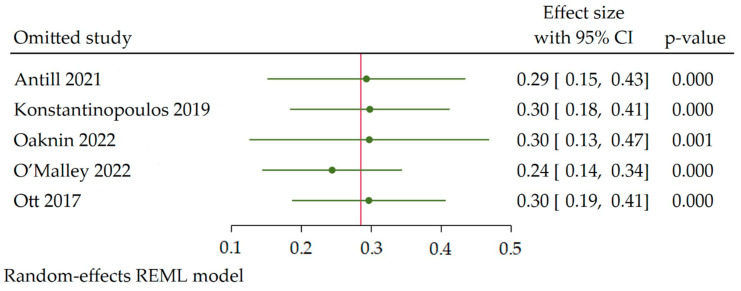
Sensitivity analysis of the meta-analysis results using the leave-one-out method [13,14,15,16,17].

**Table 1 cancers-15-04032-t001:** Risk of bias of included studies based on Newcastle–Ottawa Scale (NOS).

Authors	Domains			Results	
	Selection	Comparability	Outcome	Score	Risk
Antill 2021 [13]	****	**	*	7	Low
Konstantinopoulos 2019 [14]	****	**	***	9	Low
Oaknin 2022 [15]	****	**	**	8	Low
O’Malley 2022 [16]	****	*	*	6	Low
Ott 2017 [17]	****	*	*	6	Low

The NOS assessment contains nine stars of domains divided into three categories: selection, comparability, and outcome. Each star will be given a score of 1 (**** = score of 4; *** = score of 3; ** = score of 2; * = score of 1). Score of ≥6 indicates the study is low risk of bias.

**Table 2 cancers-15-04032-t002:** Summary of five studies of endometrial cancer patients undergoing PD-1/PD-L1 inhibitor immunotherapy.

Author/Year	Country	Study Registration no./ Name of Drug/Anti PD-1 or PD-L1	Study Phase	No. of Sample	MMR Status (n)	Objective Response Rate (ORR)		Disease Control Rate (DCR)		Adverse Event Incidence		Median PFS (Month)	Median OS (Month)
						Overall	MMR	Overall	MMR	Overall	Grade 3 or Higher		
Antill 2021 [13]	Australia	ANZGOG1601, ACTRN12617000106336, and NCT03015129/ durvalumab/ anti-PD-L1	Phase II	71	dMMR = 35	18/71	dMMR= 17/36	25/71	dMMR= 18/36	NA	NA	dMMR = 8.3	dMMR = not reached
					pMMR = 36		pMMR= 1/35		pMMR= 7/35			pMMR = 1.8	pMMR = 12.1
Konstantinopoulos 2019 [14]	USA	NCT02912572/avelumab/ anti-PD-L1	Phase II	31	dMMR = 15	5/31	dMMR= 4/15	13/31	dMMR= 8/15	22/31	6/31	dMMR = 4.4	dMMR = not reached
					pMMR = 16		pMMR= 1/16		pMMR= 5/16			pMMR = 1.9	pMMR = 6.6
Oaknin 2022 [15]	Spain	NCT02715284/ dostarlimab/ anti-PD-1	Phase I	264	dMMR = 106	69/264	dMMR= 46/106	114/264	dMMR= 59/106	196/290	48/920	NA	NA
					pMMR = 142		pMMR= 19/142		pMMR= 19/142				
O’Malley 2022 [16]	USA	NCT02628067/pembrolizumab/anti-PD-1	Phase II	90	dMMR = 90	38/79	dMMR= 38/79	NA	NA	68/90	11/90	Overall = 13.1	Overall = not reached
Ott 2017 [17]	USA	NCT02054806/pembrolizumab/anti-PD-1	Phase Ib	24	NA	3/23	NA	NA	NA	13/24	4/24	Overall = 1.8	Overall = not reached

Abbreviations: dMMR = deficient mismatch repair; pMMR = proficient mismatch repair; PFS = progression-free survival; OS = overall survival; NA = data not available.

**Table 3 cancers-15-04032-t003:** Summary of treatment-related adverse events observed in patients who received PD-1/PD-L1 inhibitor immunotherapy.

Adverse Events	Konstantinopoulos, 2019 [14]		Oaknin, 2022 [15]		O’Malley, 2022 [16]		Ott, 2018 [17]		Overall AEs, n = 435	
	Any Grade (n = 31)	Grade 3 or Higher (n = 31)	Any Grade (n = 290)	Grade 3 or Higher (n = 290)	Any Grade (n = 90)	Grade 3 or Higher (n = 90)	Any Grade (n = 24)	Grade 3 or Higher (n = 24)	Any Grade (%)	Grade 3 or Higher (%)
Anemia	3	2	27	5	22	-	-	1	11.95	1.84
Diarrhea	3	3	40	4	14	-	-	1	13.10	1.84
Fatigue	11	-	51	4	19	-	5	-	19.77	0.92
Hypothyroidism	4	1	25	-	12	-	-	-	9.43	0.23
Nausea	5	-	40	2	13	-	-	-	13.33	0.46
Arthralgia	-	-	21	-	13	-	-	-	7.82	-
Decreased appetite	-	-	18	-	8	-	3	-	6.67	-
Asthenia	-	-	31	-	-	-	-	1	7.13	0.23
Rash	-	-	21	-	10	1	-	-	7.13	0.23
Vomiting	-	-	22	-	5	-	-	-	6.21	-
Increased AST	-	-	19	3	-	-	-	-	4.37	0.69
Hypergycemia	-	-	-	3	-	-	-	1	-	0.92
Sinus bradycardia	1	1	-	-	-	-	-	-	0.23	0.23
Decreased neutrophil count	4	-	-	-	-	-	-	-	0.92	-
Myositis	1	1	-	-	-	-	-	-	0.23	0.23
Rash acneiform	2	1	-	-	-	-	-	-	0.46	0.23
Pruritus	-	-	18	-	-	-	4	-	5.06	-
Increased ALT	-	-	18	-	-	-	-	-	4.14	-
Increased amylase	-	-	16	4	-	-	-	-	3.68	0.92
Increased lipase	-	-	-	4	-	-	-	-	-	0.92
Colitis	-	-	-	2	-	-	-	-	-	0.46
Constipation	-	-	-	2	-	-	-	-	-	0.46
Hypertension	-	-	-	2	-	-	-	-	-	0.46
Pulmonary embolism	-	-	-	2	-	-	-	-	-	0.46
Dry mouth	-	-	-	-	6	-	-	-	1.38	-
Hyperthyroidism	-	-	-	-	6	-	-	-	1.38	-
Myalgia	-	-	-	-	6	-	-	-	1.38	-
Maculopapular rash	-	-	-	-	5	-	-	-	1.15	-
Increased aspartate aminotransferase	-	-	-	-	5	-	-	-	1.15	-
Pyrexia	-	-	-	-	-	-	3	1	0.69	0.23
Back pain	-	-	-	-	-	-	-	1	-	0.23
Hyponatremia	-	-	-	-	-	-	-	1	-	0.23
Chills	-	-	-	-	-	-	-	1	-	0.23

**Table 4 cancers-15-04032-t004:** Robustness check of the meta-analysis results using the leave-one-out method.

Analysis	Overall ES	Leave-One-Out Result			
		Lowest Study Range	*p*-Value	Highest Study Range	*p*-Value
Efficacy					
Overall ORR	0.26	0.14–0.34	0.000	0.13–0.47	0.001
ORR with dMMR	0.44	0.39–0.52	0.000	0.33–0.54	0.000
ORR with pMMR	0.08	0.05–0.14	0.000	−0.03–0.10	0.262
Association of ORR (dMMR vs. pMMR)	6.30	2.85–7.08	0.000	−20.27–33.62	0.627
Overall DCR	0.41	0.36–0.47	0.000	0.28–0.46	0.000
DCR with dMMR	0.54	0.38–0.71	0.000	0.24–0.78	0.000
DCR with pMMR	0.31	0.17–0.47	0.000	−0.04–0.51	0.089
Association of DCR (dMMR vs. pMMR)	2.57	1.11–3.53	0.000	−0.34–6.88	0.076
Safety					
Overall incidence of AEs	0.69	0.64–0.73	0.000	0.64–0.79	0.000
AEs incidence in grade 3 or higher	0.16	0.12–0.19	0.000	0.08–0.20	0.000

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
