# Peer review of "Efficacy and Safety of PD-1/PD-L1 Inhibitor as Single-Agent Immunotherapy in Endometrial Cancer: A Systematic Review and Meta-Analysis"

_cancers, 2023, doi:10.3390/cancers15164032_

Round 1

Reviewer 1 Report

In my opinion, the analyzed topic is interesting enough to attract the readers’ attention. I think that the abstract of this article is very clear and well structured.

In my opinion, the discussion could be studied in depth and extended. Maybe, it could be useful the evaluation of the newest insights of what is currently known about endometrial cancer diagnosis and treatment. In particular, I suggest these articles to get deeper in the topic: PMID: 36833105 and PMID: 36979434. Because of these reasons, the article should be revised and completed. Considered all these points, I think it could be of interest for the readers and, in my opinion, it deserves the priority to be published after revisions.

minor editing is necessary.

Reviewer 2 Report

In the submitted manuscript authors performed meta-analysis and systematic review to find that endometrial cancer (EC) patients with deficiency of MMR are more suitable candidates for treatment with PD-1 or PD-L1 inhibitor.

Although robust with properly applied statistical methodologies, the biggest drawback of this study is that authors didn't differentiate between studies using PD-1 or PD-L1 inhibitors. In essence, in the manuscript it is often written like those are "synonyms". Since those two types of ICIs can show different toxicity profiles and biochemical efficacy, it would be very interesting to see also if there is a difference in their effects in EC.

There are also some minor drawbacks:

1) All abbreviations mentioned in 'Abstract' should be explained (e.g., OR, CI, etc.).

2) In the first sentence of 'Introduction' it is unclear to what region do those numbers relate to, World, developed countries, USA, Malaysia or ?!

3) You should provide web addresses for all used on-line literature databases.

4) On Figure 1, I believe that in the upper right rectangle, the number of excluded records were 380, not 352.

5) In tables, whenever you note a study you should also cite the reference of that study, e.g., Antill 2021 [13].

6) All abbreviations mentioned in tables should be explained in table footnote, e.g., dMMR and pMMR in Table 2.

7) Whenever you state OR in the text or table you should also provide its p-value, 95% CI is not enough.

8) In Table 3, in the last two columns "(%)" should be put under "Any grade" and "Grade 3 or higher", not after "435".

9) p-value of 0.000 is impossible so such low p-values should be always written as "<0.001".

There are two minor drawbacks regarding English language:

- Part of sentence on page 2 "with advance stage disease had a significantly survival rate, recorded as only 18.4%" is unclear since it is unclear what "a significantly survival rate" actually means.

- There is no need to capitalize first letters in "estrogen receptor", "random effects model", "quality effects model", etc.

Round 2

Reviewer 1 Report

I read with great interest the Manuscript titled “Efficacy and Safety of PD-1/PD-L1 Inhibitor as Single Agent Immunotherapy in Endometrial Cancer: A Systematic Review and Meta-Analysis”, topic interesting enough to attract readers' attention

The quality of the manuscript has improved thanks to the changes made. I think it could be of interest to the readers and, in my opinion, it deserves the priority to be published.

Reviewer 2 Report

Authors have satisfactorily answered and addressed all my concerns, and further improved quality of this manuscript.